# Effects of Short-Term Human Albumin Infusion for the Prevention and Treatment of Hyponatremia in Patients with Liver Cirrhosis

**DOI:** 10.3390/jcm12010107

**Published:** 2022-12-23

**Authors:** Zhaohui Bai, Wentao Xu, Lu Chai, Xiaojie Zheng, Nahum Méndez-Sánchez, Cyriac Abby Philips, Gang Cheng, Xingshun Qi

**Affiliations:** 1Liver Cirrhosis Study Group, Department of Gastroenterology, General Hospital of Northern Theater Command, Shenyang 110840, China; 2NMPA Key Laboratory for Research and Evaluation of Drug Regulatory Technology, Shenyang Pharmaceutical University, Shenyang 110016, China; 3Medica Sur Clinic, National Autonomous University of Mexico, Mexico City 14050, Mexico; 4Clinical and Translational Hepatology, The Liver Institute, Center of Excellence in GI Sciences, Rajagiri Hospital, Aluva 683112, India

**Keywords:** liver cirrhosis, hyponatremia, human albumin, prevention, treatment

## Abstract

**Background**: Human albumin (HA) infusion is potentially effective for the management of hyponatremia in liver cirrhosis, but the current evidence is very limited. **Methods**: In this retrospective study, 2414 cirrhotic patients who were consecutively admitted to our hospital between January 2010 and June 2014 were included in the *Hospitalization outcome* cohort, and 339 cirrhotic patients without malignancy who were consecutively admitted to our department between December 2014 and April 2021 were included in the *Long-term outcome* cohort. The development and improvement of hyponatremia were compared between patients who received HA infusion during hospitalizations and did not. Logistic and Cox regression analyses were performed to evaluate the association of development and improvement of hyponatremia during hospitalizations with the outcomes. Odds ratios (ORs) and hazard ratios (HRs) were calculated. **Results**: In the two cohorts, HA infusion significantly decreased the incidence of hyponatremia and increased the rate of improvement of hyponatremia in cirrhotic patients during hospitalizations. In the *Hospitalization outcome* cohort, the development of hyponatremia during hospitalizations was significantly associated with increased in-hospital mortality (OR = 2.493, *p* < 0.001), and the improvement of hyponatremia during hospitalizations was significantly associated with decreased in-hospital mortality (OR = 0.599, *p* = 0.014). In the *Long-term outcome* cohort, the development of hyponatremia during hospitalizations was significantly associated with decreased long-term survival (HR = 0.400, *p* < 0.001), and the improvement of hyponatremia during hospitalizations was not significantly associated with long-term survival (HR = 1.085, *p* = 0.813). **Conclusions**: HA infusion can effectively prevent the development of hyponatremia and improve hyponatremia in cirrhotic patients during hospitalizations, which may influence the patients’ outcomes.

## 1. Introduction

Hyponatremia is the most common electrolyte disorder in liver cirrhosis [1]. It has been reported that 49.40%, 21.60%, and 1.20% of patients with liver cirrhosis and ascites have a serum sodium level of <135, 130, and 120 mmol/L, respectively [2]. Hyponatremia is associated with increased morbidity and mortality [3,4]. Notably, serum sodium level has been incorporated in the model for end-stage liver disease (MELD) score to determine the priority of liver transplantation [5]. Correction of hyponatremia can improve the cognitive function and quality of life in patients with liver cirrhosis [6,7], but its benefits on the prognosis remain unclear.

Until now, the treatment of hyponatremia in liver cirrhosis remains a clinical challenge [1]. Fluid restriction is often ineffective [8], diuretics withdrawal can worsen the severity of ascites [1], hypertonic saline is reserved for patients with severe hyponatremia and its secondary potentially life-threatening complications [9,10], and vaptans have not been sufficiently approved in clinical practice [9,10]. Human albumin (HA), which has been recommended to manage hepatorenal syndrome, spontaneous bacterial peritonitis, and large volume paracentesis [11,12,13], seems to be effective for the management of hyponatremia. However, the recommendations are heterogeneous among the current guidelines [9,10,14,15], primarily due to the lack of relevant evidence. To the best of our knowledge, no study has specifically explored the role of HA infusion on the prevention of hyponatremia in patients with liver cirrhosis, and only four cohort studies [16,17,18,19] have evaluated the role of HA infusion on the treatment of hyponatremia in patients with liver cirrhosis. Notably, among these published studies, the study design, severity of liver cirrhosis, and outcome assessment are heterogeneous.

For these reasons, our current study has three-fold objectives: (1) to clarify whether HA infusion could prevent the development of hyponatremia in liver cirrhosis; (2) to evaluate whether HA infusion could improve the severity of hyponatremia in liver cirrhosis; and (3) to explore whether the development and improvement of hyponatremia could influence the short- and long-term outcomes of patients with liver cirrhosis.

## 2. Methods

### 2.1. Study Design

This retrospective observational study was approved by the Medical Ethical Committee of the General Hospital of Northern Theater Command. The ethical approval number is Y2022-087. It includes two parts (i.e., *Hospitalization outcome* cohort and *Long-term outcome* cohort). In the *Hospitalization outcome* cohort, potentially eligible patients were screened from our retrospective database where all patients with a diagnosis of liver cirrhosis consecutively admitted to our hospital from January 2010 to June 2014 were enrolled and their outcomes during hospitalizations were observed [20]. In the *Long-term outcome* cohort, potentially eligible patients were screened from our prospective database where all patients with a diagnosis of liver cirrhosis and without malignancy consecutively admitted to the Department of Gastroenterology of our hospital from December 2014 to April 2021 were enrolled and their outcomes during follow-up were observed [21]. If serum sodium level was measured at least twice during hospitalizations, the patients would be considered in the current study.

Liver cirrhosis was diagnosed based on disease history, laboratory tests, endoscopic findings, ultrasonographic findings, and liver histology, if available. Hyponatremia was defined as a serum sodium level of <135 mmol/L, and the severity of hyponatremia was classified as mild (135–130 mmol/L), moderate (130–125 mmol/L), and severe (<125 mmol/L) [9,10]. Hyponatremia at admission was defined as the first serum sodium level measured at admission was <135 mmol/L. Hyponatremia during hospitalizations was defined as the first serum sodium level measured at admission was within the reference range (i.e., 135–145 mmol/L), but the serum sodium level rechecked during hospitalizations was <135 mmol/L. As mentioned in our previous study [20], HA was prescribed at the discretion of attending physicians, and its primary indications mainly included post-paracentesis, ascites, and hypoalbuminemia. Based on the current practice guideline, the treatments of hyponatremia mainly included water restriction, withdrawal of diuretics, hypertonic saline, and tolvaptan [22].

The data were collected regarding demographics (i.e., age and sex), etiology of liver cirrhosis (i.e., hepatitis B virus (HBV), hepatitis C virus (HCV), and alcohol), regular laboratory data (i.e., hemoglobin (Hb), white blood cell (WBC), platelet (PLT), total bilirubin (TBIL), albumin (ALB), alanine aminotransferase (ALT), alkaline phosphatase (AKP), serum creatinine (Scr), sodium (Na), potassium (K), prothrombin time (PT), and international normalized ratio (INR)). The patients’ conditions (i.e., hepatocellular carcinoma (HCC), hypokalemia, acute upper gastrointestinal bleeding (AUGIB), infection, ascites, and paracentesis) and drugs (i.e., desmopressin, terlipressin, furosemide, torasemide, spironolactone, hydrochlorothiazide, bumetanide, hypertonic saline, tolvaptan, and K supplement) that may affect serum sodium level were collected. The use of HA infusion and its dosage were also collected. All-cause death was recorded. Child–Pugh and MELD scores [23] were calculated.

### 2.2. Prevention of Hyponatremia

When the role of HA infusion for the prevention of hyponatremia was explored, the patients who underwent hemodialysis during hospitalizations or were diagnosed with hypernatremia or hyponatremia at admission were further excluded. Eligible patients assigned to the HA group should have received HA infusion before the development of hyponatremia or the last measurement of serum sodium level during hospitalizations. Otherwise, the remaining eligible patients were assigned to the control group. The development of hyponatremia was the outcome of interest as well as death. The development of hyponatremia was defined as hyponatremia was not observed at admission, but hyponatremia developed during hospitalizations.

### 2.3. Treatment of Hyponatremia

When the role of HA infusion for the treatment of hyponatremia was explored, the patients who underwent hemodialysis during hospitalizations, were diagnosed with hypernatremia at admission, or did not recheck serum sodium level after diagnosis of hyponatremia were further excluded. Eligible patients assigned to the HA group should have received HA infusion during the period from the diagnosis of hyponatremia to the last measurement of serum sodium level. Otherwise, the remaining eligible patients were assigned to the control group. The improvement of hyponatremia was the outcome of interest as well as death. The improvement of hyponatremia was defined as a reduction in the severity of hyponatremia.

### 2.4. Statistical Analyses

Continuous variables were reported as mean ± standard deviation and median (range) and compared by the non-parametric Mann–Whitney U test. Categorical variables were reported as frequency (percentage) and compared by the chi-square test. A 1:1 propensity score matching (PSM) analysis was performed. The matching factors included age, sex, Child–Pugh score, MELD score, hypokalemia, AUGIB, infection, ascites, paracentesis, desmopressin, terlipressin, furosemide, torasemide, spironolactone, hydrochlorothiazide, bumetanide, tolvaptan, hypertonic saline, and K supplement. Logistic regression analyses were conducted to explore the relationships of HA infusion with the development/improvement of hyponatremia and the effects of the development/improvement of hyponatremia on in-hospital death. Odds ratios (ORs) and 95% confidence intervals (CIs) were calculated. Cox regression analyses were also performed to explore the effects of the development/improvement of hyponatremia on long-term survival. Hazard ratios (HRs) and 95% CIs were calculated. Subgroup analyses were performed according to the presence of HCC and ascites and the use of paracentesis, if possible. In the *Long-term outcome* cohort, Kaplan–Meier curves were further drawn to demonstrate the cumulative survival and compared by the Log-rank test, and subgroup analyses were performed according to the use of HA infusion. A two-tailed *p* < 0.05 was considered statistically significant. All statistical analyses were performed with IBM SPSS 20.0 (IBM Crop, Armonk, NY, USA) software, Stata/SE 12.0 (Stata Corp, College Station, TX, USA) software, and GraphPad Prism 8.0 (GraphPad Software Inc., San Diego, CA, USA) software.

## 3. Results

### 3.1. Hospitalization Outcome Cohort

*Patients.* Overall, 4217 patients were screened, of whom 2414 were included in the *Hospitalization outcome* cohort (Figure 1A). Among them, 618 patients had hyponatremia at admission, 560 patients developed hyponatremia during hospitalizations, and 1236 patients had normal serum sodium level both at admission and during hospitalizations.

*Prevention of hyponatremia.* Overall, 1796 patients had normal serum sodium level at admission. Among them, 621 and 1175 patients were assigned to the HA and control groups, respectively. After PSM, 602 patients were included. Median total dosage of HA was 30 g (range: 10–530) in the HA group. The HA group had a significantly lower incidence of hyponatremia than the control group (16.30% versus 41.90%, *p* < 0.001) (Table 1). Similarly, logistic regression analysis also showed that HA infusion was significantly associated with decreased risk of developing hyponatremia during hospitalizations (OR = 0.270, 95% CI = 0.184–0.396, *p* < 0.001) (Figure 2A).

Regardless of HCC, ascites, and paracentesis, the HA group had a significantly lower incidence of hyponatremia than the control group; and logistic regression analyses also showed that HA infusion was significantly associated with decreased risk of developing hyponatremia during hospitalizations (Figure 2A).

Five hundred and sixty patients developed hyponatremia during hospitalizations and 1236 did not. Among them, 77 patients died during hospitalizations. Causes of death were related (*n* = 65) and unrelated (*n* = 12) to liver diseases. Patients who developed hyponatremia during hospitalizations had a significantly higher in-hospital mortality than those who did not (7.10% versus 3.00%, *p* < 0.001). Results remained in both HA (10.40% versus 4.30%, *p* = 0.004) and control (5.60% versus 2.30%, *p* = 0.003) groups. Similarly, logistic regression analysis also showed that the development of hyponatremia during hospitalizations was significantly associated with increased in-hospital mortality (OR = 2.493, 95% CI = 1.576–3.944, *p* < 0.001). Results remained in both HA (OR = 2.555, 95% CI = 1.319–4.948, *p* = 0.005) and control (OR = 2.556, 95% CI = 1.345–4.857, *p* = 0.004) groups.

*Treatment of hyponatremia.* Overall, 1178 patients were diagnosed with hyponatremia at admission/during hospitalizations. Among them, 174 patients who did not recheck serum sodium level after the diagnosis of hyponatremia were excluded. Finally, 1004 patients were included. Among them, 545 and 459 patients were assigned to the HA and control groups, respectively. After PSM, 394 patients were included. Median total dosage of HA was 40 g (range: 10–380) in the HA group. The HA group had a significantly higher rate of improvement of hyponatremia than the control group (82.70% versus 54.80%, *p* < 0.001) (Table 2). Similarly, logistic regression analysis showed that HA infusion was significantly associated with increased rate of improvement of hyponatremia during hospitalizations (OR = 3.951, 95% CI = 2.484–6.283, *p* < 0.001) (Figure 2B).

Regardless of HCC and ascites, the HA group had a significantly higher rate of improvement of hyponatremia than the control group. This difference remained significant in patients who did not undergo paracentesis, but not in those who underwent paracentesis. Logistic regression analyses showed that HA infusion was significantly associated with increased rate of improvement of hyponatremia during hospitalizations in patients with HCC, non-HCC, ascites, and who did not undergo paracentesis, but not those without ascites or who underwent paracentesis (Figure 2B).

Six hundred and forty-two patients had improvement of hyponatremia during hospitalizations and 362 did not. Among them, 104 patients died during hospitalizations. Causes of death were related (*n* = 88) and unrelated (*n* = 16) to liver diseases. Patients who had improvement of hyponatremia during hospitalizations had a significantly lower in-hospital mortality than those who did not (8.60% versus 13.50%, *p* = 0.013). Results remained in control group (6.70% versus 14.20%, *p* = 0.008), but not in HA group (10.00% versus 12.90%, *p* = 0.309). Similarly, logistic regression analysis also showed that the improvement of hyponatremia during hospitalizations was significantly associated with decreased in-hospital mortality (OR = 0.599, 95% CI = 0.398–0.901, *p* = 0.014). Results remained in control group (OR = 0.435, 95% CI = 0.232–0.815, *p* = 0.009), but not in HA group (OR = 0.752, 95% CI = 0.434–1.304, *p* = 0.310).

### 3.2. Long-Term Outcome Cohort

*Patients*. Overall, 544 patients were screened, of whom 339 were included in the *Long-term outcome* cohort (Figure 1B). Among them, 48 patients had hyponatremia at admission, 61 patients developed hyponatremia during hospitalizations, and 230 patients had normal serum sodium level both at admission and during hospitalizations.

*Prevention of hyponatremia.* Overall, 291 patients had normal serum sodium level at admission. Among them, 93 and 198 patients were assigned to the HA and control groups, respectively. After PSM, 78 patients were included. Median total dosage of HA was 30 g (range: 10–150) in the HA group. The HA group had a significantly lower incidence of hyponatremia than the control group (7.70% versus 30.80%, *p* = 0.010) (Table 3). Similarly, logistic regression analysis also showed that HA infusion was significantly associated with decreased risk of developing hyponatremia during hospitalizations (OR = 0.188, 95% CI = 0.048–0.731, *p* = 0.016).

Sixty-one patients developed hyponatremia during hospitalizations and 230 did not. During a median follow-up period of 37.12 months (range: 0.30–82.55), 97 patients died. Causes of death were related (*n* = 70) and unrelated (*n* = 27) to liver diseases. Cox regression analysis demonstrated that the development of hyponatremia during hospitalizations was significantly associated with decreased long-term survival (HR = 0.400, 95% CI = 0.260–0.616, *p* < 0.001). Results remained in both HA (HR = 0.460, 95% CI = 0.250–0.845, *p* = 0.012) and control (HR = 0.380, 95% CI = 0.208–0.697, *p* = 0.002) groups. Kaplan–Meier curve analysis also showed that patients who developed hyponatremia during hospitalizations had a significantly lower cumulative survival rate than those who did not (Log-rank test: *p* < 0.001) (Figure 3A). Results remained in both HA (Log-rank test: *p* = 0.010) (Figure 3B) and control (Log-rank test: *p* = 0.001) (Figure 3C) groups.

*Treatment of hyponatremia*. Overall, 109 patients were diagnosed with hyponatremia at admission/during hospitalizations. Among them, 21 patients who did not recheck serum sodium level after the diagnosis of hyponatremia were excluded. Finally, 88 patients were included. Among them, 42 and 46 patients were assigned to the HA and control groups, respectively. After PSM, 16 patients were included. Median total dosage of HA was 40 g (range: 20–180) in the HA group. The HA group had a significantly higher rate of improvement of hyponatremia than the control group (87.50% versus 37.50%, *p* = 0.039) (Table 4). Logistic regression analysis showed that HA infusion was not significantly associated with increased improvement of hyponatremia during hospitalizations (OR = 11.667, 95% CI = 0.922–147.563, *p* = 0.058).

Fifty-nine patients had improvement of hyponatremia during hospitalizations and 29 did not. During a median follow-up period of 30.72 months (range: 0.21–76.18), 41 patients died. Causes of death were related (*n* = 37) and unrelated (*n* = 4) to liver diseases. Cox regression analysis demonstrated that the improvement of hyponatremia during hospitalizations was not significantly associated with increased long-term survival (HR = 1.085, 95% CI = 0.553–2.127, *p* = 0.813). Results remained in both HA (HR = 1.352, 95% CI = 0.523–3.495, *p* = 0.534) and control (HR = 0.864, 95% CI = 0.325–2.297, *p* = 0.769) groups. Kaplan–Meier curve analysis also demonstrated that the cumulative survival was not significantly different between patients who had improvement of hyponatremia during hospitalizations and those who did not (Log-rank test: *p* = 0.813) (Figure 4A). Results remained in both HA (Log-rank test: *p* = 0.533) (Figure 4B) and control (Log-rank test: *p* = 0.596) (Figure 4C) groups.

## 4. Discussion

Our study has two major findings: (1) HA infusion can effectively reduce the incidence of hyponatremia during hospitalizations in patients with liver cirrhosis and normal serum sodium level at admission, and the development of hyponatremia during hospitalizations can worsen both in-hospital and long-term outcomes; and (2) HA infusion can effectively improve serum sodium level during hospitalizations in patients with liver cirrhosis and hyponatremia, and the improvement of hyponatremia should be beneficial for in-hospital outcome, but not for long-term outcome.

Hypervolemic hyponatremia is the most common type of hyponatremia in patients with liver cirrhosis, accounting for more than 90% [24]. It is mainly related to water retention secondary to increased secretion of antidiuretic hormone [25], which is caused by splanchnic vasodilation associated with portal hypertension, systemic inflammation, and hyperdynamic circulation in advanced liver cirrhosis [26,27]. HA can bind to endogenous and exogenous compounds, thereby exerting antioxidant activity, modulating inflammation and immune responses, improving cardiac function, and restoring endothelial integrity [12,28,29]. Therefore, HA infusion may be theoretically appropriate for the management of hyponatremia in patients with liver cirrhosis.

To the best of our knowledge, only four studies have explored the role of HA infusion for the treatment of hyponatremia [16,17,18,19]. In 2017, Shen et al.’s cohort study, which included 146 patients with hyponatremia, showed that a change of serum sodium level was similar between HA and crystalloid groups, and that HA infusion was associated with reduced 6-month mortality [16]. In 2018, Bajaj et al.’s cohort study, which included 1126 patients with hyponatremia, showed that HA group had a significantly higher rate of hyponatremia resolution, but a higher 30-day mortality than control group [17]. In 2021, China et al.’s post hoc analysis of ATTIRE trial, which included 206 patients with hyponatremia, showed that HA group had a significantly higher serum sodium level than control group [18]. In 2022, Zaccherini et al.’s post hoc analysis of ANSWER study, which included 431 patients with hyponatremia, showed that HA infusion can improve hyponatremia and reduce episodes of at least moderate hyponatremia in outpatients with cirrhosis and ascites [19].

As compared to these previous studies, our current study has some strengths in terms of study design. First, the severity of liver cirrhosis may affect the efficacy of HA infusion in liver cirrhosis. In our study, PSM analyses were employed to balance the severity of liver cirrhosis between patients who received and did not receive HA. By comparison, in Shen et al.’s study [16], the HA group had significantly higher proportions of ascites, refractory ascites, and diuretic use and higher MELD score than the control group. In Bajaj et al.’s study [17], the HA group had significantly higher proportions of infection, spontaneous bacterial peritonitis, renal dysfunction, large volume paracentesis, and organ failure and higher Child–Pugh and MELD scores than the control group. Baseline characteristics of patients with hyponatremia were not clearly reported in China et al.’s [18] and Zaccherini et al.’s [19] studies. Second, HA was selectively infused in the control group in Shen et al.’s [16], China et al.’s [18], and Zaccherini et al.’s [19] studies, which may cause a bias in assessing the outcomes. By comparison, none received HA infusion in the control group in our study. Third, HA was infused at a median dosage of 225 g in Bajaj et al.’s [17] study during hospitalizations and a mean dosage of 239.4 g in China et al.’s [18] study during a 14-day period. It is more likely that high-dose HA infusion can cause serious adverse events, such as pulmonary edema [30]. By comparison, only a relatively low dosage of HA infusion (median: 40 g) during hospitalizations was employed in our study, which is similar to that in the ANSWER study [31]. Fourth, ascites and HCC are common predisposing factors of hyponatremia in patients with liver cirrhosis [32,33,34], and hyponatremia also significantly worsens the outcomes of cirrhotic patients with ascites [4,35] and HCC [36,37]. Thus, our study conducted subgroup analyses to evaluate the efficacy of HA infusion for correction of hyponatremia according to the presence of ascites and HCC, which have not been performed in previous studies yet.

No previous study has specifically explored the role of HA infusion for the prevention of hyponatremia in patients with liver cirrhosis. However, some studies, which primarily evaluated the efficacy of HA infusion for the prevention of post-paracentesis circulatory dysfunction in cirrhotic patients with ascites undergoing large volume paracentesis, reported that HA infusion could significantly decrease the incidence of hyponatremia after large volume paracentesis [38,39]. By comparison, our study has for the first time demonstrated that HA infusion may prevent from hyponatremia in general patients with liver cirrhosis during hospitalizations.

As known, hyponatremia is an important prognostic factor of patients with liver cirrhosis [3,40,41], because it can predispose to more severe complications [42,43,44]. Our study further confirmed that the development of hyponatremia significantly increased the risk of in-hospital and long-term death. By comparison, few studies explored the impact of the improvement of hyponatremia on the prognosis of liver cirrhosis. Until now, only a previous cohort study demonstrated the benefits of the improvement of hyponatremia by tolvaptan on the short-term survival of patients with liver cirrhosis [45]. Similarly, our study found that the improvement of hyponatremia was significantly associated with a lower risk of in-hospital death, but could not support its benefit in reducing the long-term mortality. This is probably because long-term outcome may be influenced by multiple factors in patients with liver cirrhosis, especially Child–Pugh [46] and MELD [47] scores. Additionally, despite the improvement of hyponatremia during hospitalizations, the progression of liver cirrhosis and recurrence of hyponatremia during follow-up had not been evaluated in our study.

Our study has several limitations. First, the type of hyponatremia (i.e., hypervolemic, hypovolemic, or euvolemic) was not clearly identified due to a lack of data regarding patients’ volume status. However, we performed subgroup analyses according to the presence of ascites. Second, the etiology of hyponatremia could not be clarified, but some potential risk factors, including hypokalemia, AUGIB, infection, ascites, paracentesis, desmopressin, terlipressin, furosemide, torasemide, spironolactone, hydrochlorothiazide, and bumetanide, were adjusted. Third, the development and improvement of hyponatremia during follow-up were not available. Fourth, the information regarding HA infusion after discharge were not available in the *Long-term outcome* cohort, thus the role of long-term HA infusion could not be explored. Fifth, the first or recurring episodes of hyponatremia could not be clearly identified. Finally, the patient selection bias was often unavoidable due to the retrospective nature of this study.

## 5. Conclusions

HA infusion may be effective for preventing and treating hyponatremia in patients with liver cirrhosis during hospitalizations, which may be beneficial for the patients’ outcomes. Randomized controlled trials should be warranted to clarify the role of HA infusion for the management of hyponatremia in patients with liver cirrhosis.

## Figures and Tables

**Figure 1 jcm-12-00107-f001:**
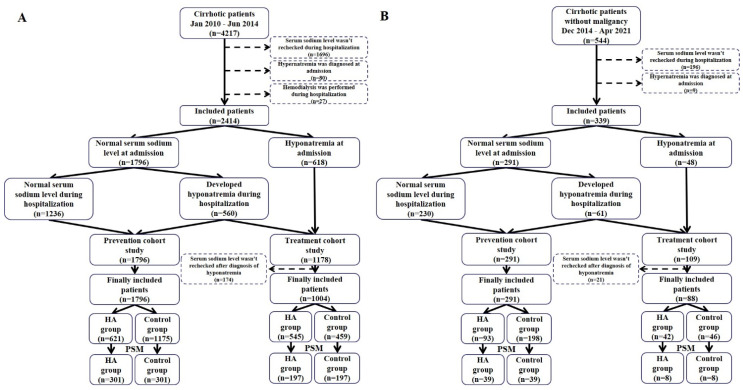
Flow charts of patient selection in the *Hospitalization outcome* (**panel A**) and *Long-term outcome* (**panel B**) cohorts. Abbreviations: PSM: propensity score matching; HA: human albumin.

**Figure 2 jcm-12-00107-f002:**
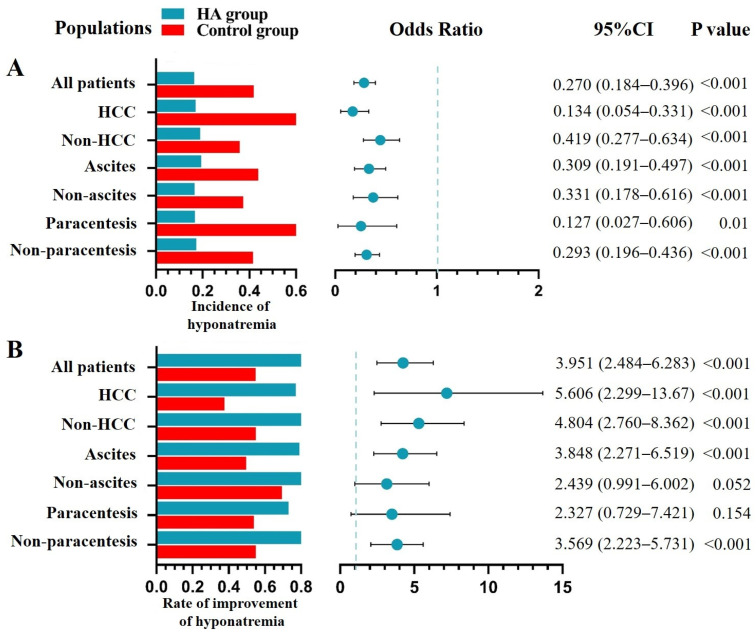
HA infusion for the prevention and treatment of hyponatremia during hospitalizations. (**Panel A**): Bar plots displayed the incidence of hyponatremia in the HA and control groups, and forest plots showed the ORs with 95% CIs for the role of HA infusion for the prevention of hyponatremia. (**Panel B**): Bar plots displayed the rate of improvement of hyponatremia in the HA and control groups, and forest plots showed the ORs with 95% CIs for the role of HA infusion for the improvement of hyponatremia. Abbreviations: HA: human albumin; CI: confidence intervals; HCC: hepatocellular carcinoma.

**Figure 3 jcm-12-00107-f003:**
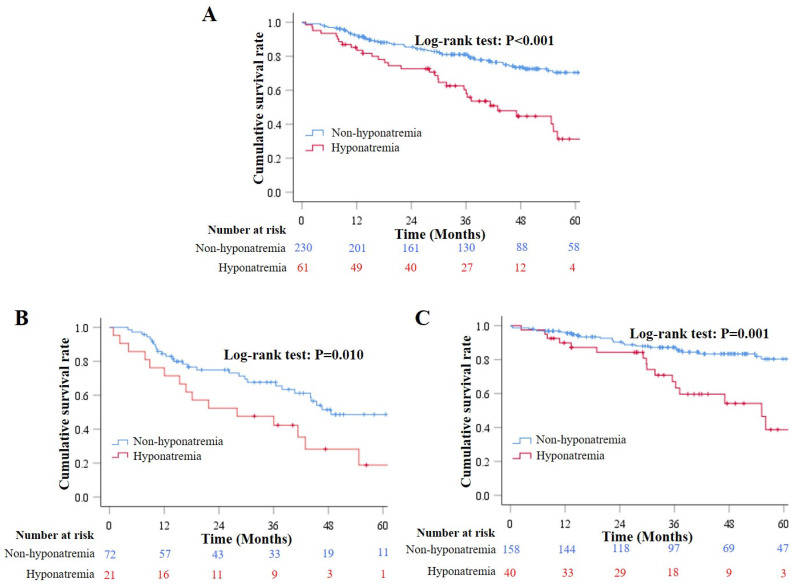
Long-term survival according to the development of hyponatremia during hospitalizations. (**Panel A**): Kaplan–Meier curves showed the cumulative survival rates in overall patients who developed and did not develop hyponatremia during hospitalizations. (**Panel B**): Kaplan–Meier curves showed the cumulative survival rates in overall patients who developed and did not develop hyponatremia during hospitalizations in the HA group. (**Panel C**): Kaplan–Meier curves showed the cumulative survival rates in overall patients who developed and did not develop hyponatremia during hospitalizations in the control group.

**Figure 4 jcm-12-00107-f004:**
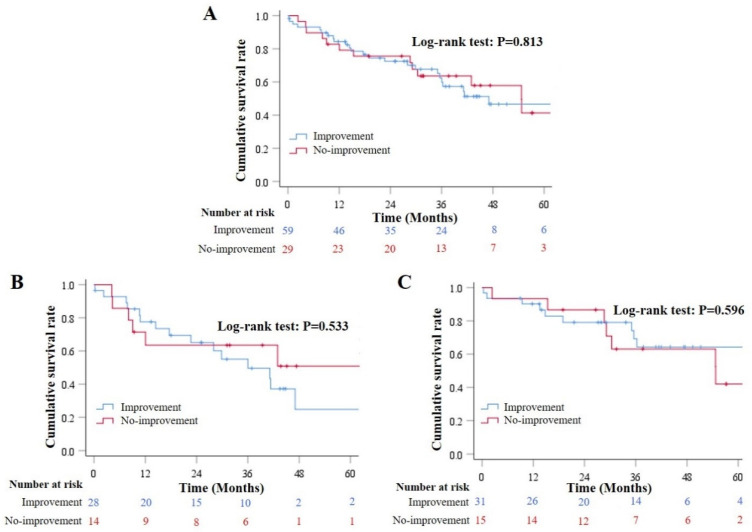
Long-term survival according to the improvement of hyponatremia during hospitalizations. (**Panel A**): Kaplan–Meier curves showed the cumulative survival rates in overall patients who had and did not have improvement of hyponatremia during hospitalizations. (**Panel B**): Kaplan–Meier curves showed the cumulative survival rates in overall patients who had and did not have improvement of hyponatremia during hospitalizations in the HA group. (**Panel C**): Kaplan–Meier curves showed the cumulative survival rates in overall patients who had and did not have improvement of hyponatremia during hospitalizations in the control group.

**Table 1 jcm-12-00107-t001:** *Hospitalization outcome* cohort—Characteristics of patients in the prevention study after PSM.

Variables	No. Pts	Overall	No. Pts	HA Group	No. Pts	Control Group	*p* Value
Age (years)	602	58.26 (21.14–87.82) 59.15 ± 11.85	301	58.46 (21.14–87.82) 59.12 ± 12.02	301	57.69 (29.66–86.00) 59.17 ± 11.69	0.916
Sex (male) (%)	602	389 (64.60%)	301	192 (63.80%)	301	197 (65.40%)	0.670
Etiology of liver cirrhosis							
HBV (%)	602	265 (44.00%)	301	145 (48.20%)	301	120 (39.90%)	0.040
HCV (%)	602	57 (9.50%)	301	33 (11.00%)	301	24 (8.00%)	0.210
Alcohol (%)	602	175 (29.10%)	301	88 (29.20%)	301	87 (28.90%)	0.928
HCC (%)	602	167 (27.70%)	301	80 (26.60%)	301	87 (28.90%)	0.524
Hypokalemia (%)	602	73 (12.10%)	301	36 (12.00%)	301	37 (12.30%)	0.901
AUGIB (%)	602	177 (29.40%)	301	88 (29.20%)	301	89 (29.60%)	0.929
Infection (%)	602	197 (32.70%)	301	95 (31.60%)	301	102 (33.90%)	0.543
Ascites (%)	602	349 (58.00%)	301	176 (58.50%)	301	173 (57.50%)	0.804
Paracentesis * (%)	602	43 (7.10%)	301	18 (6.00%)	301	25 (8.30%)	0.268
Laboratory tests							
Hb (g/L)	602	92.00 (27.00–169.00) 93.84 ± 28.00	301	91.00 (29.00–164.00) 91.83 ± 26.89	301	93.00 (27.00–169.00) 95.84 ± 28.97	0.132
WBC (10^9^/L)	602	4.10 (0.50–33.50) 5.18 ± 3.94	301	4.00 (0.90–30.20) 4.96 ± 3.61	301	4.20 (0.50–33.50) 5.41 ± 4.24	0.342
PLT (10^9^/L)	602	76.50 (9.00–775.00) 97.62 ± 81.23	301	74.00 (16.00–394.00) 89.86 ± 57.31	301	78.00 (9.00–775.00) 105.37 ± 99.06	0.327
TBIL (μmol/L)	602	24.65 (3.40–576.40) 40.77 ± 59.28	301	23.40 (3.40–423.50) 34.62 ± 46.53	301	26.70 (5.10–576.40) 46.92 ± 69.27	0.048
ALB (g/L)	602	30.60 (10.00–53.90) 31.11 ± 6.56	301	29.30 (13.50–48.50) 29.75 ± 6.42	301	32.30 (10.00–53.90) 32.47 ± 6.41	<0.001
ALT (U/L)	602	27.50 (4.00–1460.00) 51.84 ± 117.83	301	27.00 (4.00–730.00) 43.86 ± 58.71	301	28.00 (6.00–1460.00) 59.82 ± 155.69	0.882
AKP (U/L)	602	92.00 (1.30–782.00) 119.85 ± 93.81	301	91.80 (7.05–782.00) 122.82 ± 107.90	301	92.20 (1.30–511.00) 116.89 ± 77.27	0.437
Scr (μmol/L)	602	61.80 (2.60–742.00) 76.22 ± 66.70	301	62.00 (24.00–742.00) 75.20 ± 58.27	301	61.00 (2.60–715.00) 77.23 ± 74.26	0.688
K (mmol/L)	602	4.05 (2.05–6.14) 4.04 ± 0.52	301	4.03 (2.65–5.57) 4.04 ± 0.49	301	4.07 (2.05–6.14) 4.04 ± 0.55	0.849
Na (mmol/L)	602	139.30 (135.00–145.00) 139.39 ± 2.57	301	139.50 (135.00–145.00) 139.50 ± 2.52	301	139.10 (135.00–145.00) 139.29 ± 2.61	0.294
PT (seconds)	602	15.80 (11.00–51.00) 16.60 ± 4.16	301	16.00 (11.00–33.70) 16.48 ± 3.29	301	15.50 (11.30–51.00) 16.71 ± 4.89	0.349
INR	602	1.27 (0.79–11.70) 1.39 ± 0.65	301	1.28 (0.79–3.28) 1.35 ± 0.36	301	1.24 (0.82–11.70) 1.43 ± 0.85	0.634
Child–Pugh score	602	8 (5–14) 7.92 ± 1.91	301	8 (5–14) 7.89 ± 1.78	301	8 (5–13) 7.96 ± 2.03	0.929
MELD score	602	6.92 (−21.42–44.70) 8.03 ± 6.76	301	6.77 (−5.62–26.86) 7.69 ± 5.90	301	7.24 (−21.42–44.70) 8.37 ± 7.51	0.561
Treatments							
Desmopressin (%)	602	46 (7.60%)	301	18 (6.00%)	301	28 (9.30%)	0.125
Terlipressin (%)	602	0	301	0	301	0	\
Furosemide (%)	602	390 (64.80%)	301	196 (65.10%)	301	194 (64.50%)	0.864
Torasemide (%)	602	249 (41.40%)	301	133 (44.20%)	301	116 (38.50%)	0.159
Spironolactone (%)	602	302 (50.20%)	301	156 (51.80%)	301	146 (48.50%)	0.415
Hydrochlorothiazide (%)	602	4 (0.70%)	301	3 (1.00%)	301	1 (0.30%)	0.316
Bumetanide (%)	602	8 (1.30%)	301	4 (1.30%)	301	4 (1.30%)	1.000
Tolvaptan (%)	602	2 (0.30%)	301	1 (0.30%)	301	1 (0.30%)	1.000
Hypertonic saline ^#^ (%)	602	10 (1.70%)	301	4 (1.30%)	301	6 (2.00%)	0.524
K supplement (%)	602	434 (72.10%)	301	219 (72.80%)	301	215 (71.40%)	0.716
HA dosage (g)	301	30 (10–530) 46.84 ± 46.42	301	30 (10–530) 46.84 ± 46.42	NA	NA	\
Incidence of hyponatremia (%)	602	175 (29.10%)	301	49 (16.30%)	301	126 (41.90%)	<0.001

Notes: * Data regarding use of paracentesis were extracted before the development of hyponatremia. ^#^ Data regarding use of hypertonic saline were extracted before the development of hyponatremia. Abbreviations: PSM: propensity score matching; Pts: patients; HA: human albumin; HBV: hepatitis B virus; HCV: hepatitis C virus; HCC: hepatocellular carcinoma; AUGIB: acute upper gastrointestinal bleeding, Hb: hemoglobin; WBC: white blood cell; PLT: platelet; TBIL: total bilirubin; ALB: albumin; ALT: alanine aminotransferase; AKP: alkaline phosphatase; Scr: serum creatinine; K: potassium; Na: sodium; PT: prothrombin time; INR: international normalized ratio; MELD: model for end-stage liver disease; NA: not applicable.

**Table 2 jcm-12-00107-t002:** *Hospitalization outcome* cohort—Characteristics of patients in the treatment study after PSM.

Variables	No. Pts	Overall	No. Pts	HA Group	No. Pts	Control Group	*p* Value
Age (years)	394	58.10 (29.94–89.19) 59.48 ± 11.52	197	58.20 (37.88–85.92) 59.76 ± 11.91	197	57.97 (29.94–89.19) 59.19 ± 11.14	0.777
Sex (male) (%)	394	280 (71.10%)	197	142 (72.10%)	197	138 (70.10%)	0.657
Etiology of liver cirrhosis							
HBV (%)	394	153 (38.80%)	197	82 (41.60%)	197	71 (36.00%)	0.256
HCV (%)	394	48 (12.20%)	197	28 (14.20%)	197	20 (10.20%)	0.218
Alcohol (%)	394	137 (34.80%)	197	66 (33.50%)	197	71 (36.00%)	0.597
HCC (%)	394	90 (22.80%)	197	51 (25.90%)	197	39 (19.80%)	0.150
Hypokalemia (%)	394	63 (16.00%)	197	33 (16.80%)	197	30 (15.20%)	0.680
AUGIB (%)	394	81 (20.60%)	197	38 (19.30%)	197	43 (21.80%)	0.533
Infection (%)	394	161 (40.90%)	197	81 (41.10%)	197	80 (40.60%)	0.918
Ascites (%)	394	270 (68.50%)	197	136 (69.00%)	197	134 (68.00%)	0.828
Paracentesis * (%)	394	51 (12.90%)	197	23 (11.70%)	197	28 (14.20%)	0.453
Severity of hyponatremia							
Mild (%)/Moderate (%)/Severe (%)	394	313 (79.40%)/58 (14.70%)/23 (5.80%)	197	155 (78.70%)/29 (14.70%)/13 (6.60%)	197	158 (80.20%)/29 (14.70%)/10 (5.10%)	0.811
Laboratory tests							
Hb (g/L)	394	94.00 (35.00–180.00) 93.93 ± 28.35	197	93.00 (36.00–180.00) 94.04 ± 29.00	197	96.00 (35.00–157.00) 93.81 ± 27.75	0.812
WBC (10^9^/L)	394	5.50 (0.50–31.10) 6.61 ± 4.64	197	5.70 (0.90–31.10) 6.88 ± 5.06	197	5.50 (0.50–30.70) 6.33 ± 4.17	0.636
PLT (10^9^/L)	394	84.00 (5.00–464.00) 100.93 ± 71.42	197	81.00 (13.00–365.00) 99.57 ± 66.64	197	84.00 (5.00–464.00) 102.28 ± 76.04	0.907
TBIL (μmol/L)	394	33.30 (2.70–809.80) 72.40 ± 107.12	197	31.10 (2.70–454.70) 62.18 ± 82.29	197	36.40 (4.20–809.80) 82.63 ± 126.71	0.155
ALB (g/L)	394	29.00 (12.40–52.80) 29.58 ± 6.58	197	27.90 (12.40–50.00) 28.89 ± 6.60	197	29.60 (13.70–52.80) 30.27 ± 6.48	0.017
ALT (U/L)	394	31.00 (7.00–3471.00) 63.59 ± 197.28	197	30.00 (7.00–3471.00) 70.23 ± 256.54	197	33.00 (8.00–1335.00) 56.94 ± 110.16	0.508
AKP (U/L)	394	104.00 (35.00–1075.00) 143.88 ± 126.34	197	100.00 (39.00–586.00) 128.25 ± 91.63	197	109.00 (35.00–1075.00) 159.51 ± 152.04	0.122
Scr (μmol/L)	394	64.00 (24.00–761.00) 88.89 ± 85.07	197	65.00 (30.00–636.00) 88.10 ± 74.03	197	63.00 (24.00–761.00) 89.69 ± 95.02	0.242
K (mmol/L)	394	4.04 (2.09–6.95) 4.07 ± 0.68	197	4.08 (2.09–6.95) 4.09 ± 0.68	197	4.02 (2.17–6.37) 4.05 ± 0.69	0.600
Na (mmol/L)	394	132.55 (102.90–134.90) 131.54 ± 3.59	197	132.60 (115.80–134.90) 131.73 ± 3.20	197	132.40 (102.90–134.90) 131.35 ± 3.94	0.381
PT (seconds)	394	16.25 (11.00–63.30) 17.58 ± 5.21	197	16.50 (11.00–63.30) 17.66 ± 5.53	197	16.00 (11.20–40.90) 17.49 ± 4.87	0.427
INR	394	1.31 (0.81–11.70) 1.51 ± 0.81	197	1.32 (0.84–8.05) 1.49 ± 0.70	197	1.30 (0.81–11.70) 1.53 ± 0.92	0.600
Child–Pugh score	394	9 (5–15) 8.82 ± 2.16	197	9 (5–14) 8.70 ± 2.03	197	9 (5–15) 8.95 ± 2.28	0.695
MELD score	394	9.58 (−5.22–43.97) 11.12 ± 8.75	197	9.70 (−5.22–43.97) 10.95 ± 8.23	197	9.39 (−4.79–40.95) 11.28 ± 9.26	0.986
Treatments							
Desmopressin (%)	394	11 (2.80%)	197	7 (3.60%)	197	4 (2.00%)	0.359
Terlipressin (%)	394	0	197	0	197	0	\
Furosemide (%)	394	294 (74.60%)	197	146 (74.10%)	197	148 (75.10%)	0.817
Torasemide (%)	394	191 (48.50%)	197	99 (50.30%)	197	92 (46.70%)	0.480
Spironolactone (%)	394	227 (57.60%)	197	114 (57.90%)	197	113 (57.40%)	0.919
Hydrochlorothiazide (%)	394	4 (1.00%)	197	2 (1.00%)	197	2 (1.00%)	1.000
Bumetanide (%)	394	9 (2.30%)	197	5 (2.50%)	197	4 (2.00%)	0.736
Tolvaptan (%)	394	0	197	0	197	0	\
Hypertonic saline ^#^ (%)	394	34 (8.60%)	197	17 (8.60%)	197	17 (8.60%)	1.000
K supplement (%)	394	291 (73.90%)	197	147 (74.60%)	197	144 (73.10%)	0.731
HA dosage (g)	197	40.00 (10.00–380.00) 53.20 ± 47.48	197	40.00 (10.00–380.00) 53.20 ± 47.48	NA	NA	\
Improvement of hyponatremia (%)	394	271 (68.80%)	197	163 (82.70%)	197	108 (54.80%)	<0.001

Notes: * Data regarding use of paracentesis were extracted after the development of hyponatremia. ^#^ Data regarding use of hypertonic saline were extracted after the development of hyponatremia. Abbreviations: PSM: propensity score matching; Pts: patients; HA: human albumin; HBV: hepatitis B virus; HCV: hepatitis C virus; HCC: hepatocellular carcinoma; AUGIB: acute upper gastrointestinal bleeding, Hb: hemoglobin; WBC: white blood cell; PLT: platelet; TBIL: total bilirubin; ALB: albumin; ALT: alanine aminotransferase; AKP: alkaline phosphatase; Scr: serum creatinine; K: potassium; Na: sodium; PT: prothrombin time; INR: international normalized ratio; MELD: model for end-stage liver disease; NA: not applicable.

**Table 3 jcm-12-00107-t003:** *Long-term outcome* cohort—Characteristics of patients in the prevention study after PSM.

Variables	No. Pts	Overall	No. Pts	HA Group	No. Pts	Control Group	*p* Value
Age (years)	78	57.93 (30.21–78.36) 57.49 ± 10.71	39	58.10 (30.21–78.36) 56.59 ± 11.40	39	58.10 (33.61–77.30) 58.39 ± 10.04	0.371
Sex (male) (%)	78	51 (65.40%)	39	26 (66.70%)	39	25 (64.10%)	0.812
Etiology of liver cirrhosis							
HBV (%)	78	32 (41.00%)	39	18 (46.20%)	39	14 (35.90%)	0.357
HCV (%)	78	4 (5.10%)	39	3 (7.70%)	39	1 (2.60%)	0.305
Alcohol (%)	78	30 (38.50%)	39	14 (35.90%)	39	16 (41.00%)	0.642
Hypokalemia (%)	78	16 (20.50%)	39	6 (15.40%)	39	10 (25.60%)	0.262
AUGIB (%)	78	27 (34.60%)	39	12 (30.80%)	39	15 (38.50%)	0.475
Infection (%)	78	8 (10.30%)	39	5 (12.80%)	39	3 (7.70%)	0.455
Ascites (%)	78	65 (83.30%)	39	32 (82.10%)	39	33 (84.60%)	0.761
Paracentesis * (%)	78	0	39	0	39	0	\
Laboratory tests							
Hb (g/L)	78	85.00 (37.00–150.00) 89.95 ± 26.53	39	95.00 (43.00–150.00) 93.13 ± 24.82	39	80.00 (37.00–136.00) 86.77 ± 28.10	0.259
WBC (10^9^/L)	78	3.70 (0.80–10.60) 4.15 ± 2.03	39	4.30 (1.00–10.60) 4.40 ± 1.96	39	3.50 (0.80–9.30) 3.90 ± 2.10	0.106
PLT (10^9^/L)	78	77.50 (19.00–470.00) 100.92 ± 71.19	39	79.00 (19.00–470.00) 105.15 ± 80.12	39	77.00 (34.00–302.00) 96.69 ± 61.74	0.649
TBIL (μmol/L)	78	27.05 (8.00–281.10) 38.08 ± 40.23	39	27.50 (8.80–100.40) 33.92 ± 24.29	39	24.70 (8.00–281.10) 42.23 ± 51.52	0.901
ALB (g/L)	78	28.70 (19.00–38.00) 28.58 ± 4.40	39	26.80 (19.00–38.00) 27.25 ± 4.62	39	30.50 (21.50–37.00) 29.91 ± 3.78	0.009
ALT (U/L)	78	26.55 (8.17–613.24) 48.72 ± 75.91	39	27.30 (12.18–241.21) 47.07 ± 43.79	39	24.95 (8.17–613.24) 50.37 ± 98.76	0.169
AKP (U/L)	78	103.97 (31.00–2525.27) 171.11 ± 299.77	39	103.66 (31.00–983.93) 143.44 ± 152.69	39	104.62 (33.66–2525.27) 198.78 ± 396.49	0.964
Scr (μmol/L)	78	60.81 (40.21–178.55) 67.55 ± 22.28	39	72.04 (40.70–178.55) 74.86 ± 26.14	39	54.45 (40.21–99.20) 60.24 ± 14.58	0.004
K (mmol/L)	78	3.84 (2.42–5.19) 3.81 ± 0.49	39	3.86 (2.42–5.19) 3.85 ± 0.53	39	3.79 (2.84–4.70) 3.77 ± 0.44	0.433
Na (mmol/L)	78	138.10 (135.50–144.30) 138.62 ± 2.18	39	137.60 (135.50–144.30) 138.33 ± 2.09	39	138.40 (135.50–144.20) 138.92 ± 2.26	0.243
PT (seconds)	78	15.95 (12.50–27.40) 16.61 ± 2.73	39	16.00 (12.60–23.90) 16.66 ± 2.35	39	15.60 (12.50–27.40) 16.56 ± 3.10	0.330
INR	78	1.27 (0.94–2.55) 1.36 ± 0.28	39	1.30 (1.00–2.08) 1.37 ± 0.24	39	1.26 (0.94–2.55) 1.36 ± 0.33	0.298
Child–Pugh score	78	8 (5–12) 8.33 ± 1.30	39	8 (6–12) 8.41 ± 1.29	39	8 (5–11) 8.26 ± 1.31	0.905
MELD score	78	8.05 (−2.35–22.73) 8.22 ± 4.90	39	8.50 (−2.35–19.43) 9.18 ± 4.68	39	7.08 (−0.56–22.73) 7.26 ± 4.99	0.055
Treatments							
Desmopressin (%)	78	0	39	0	39	0	\
Terlipressin (%)	78	9 (11.50%)	39	5 (12.80%)	39	4 (10.30%)	0.723
Furosemide (%)	78	28 (35.90%)	39	13 (33.33%)	39	15 (38.50%)	0.637
Torasemide (%)	78	38 (48.70%)	39	20 (51.30%)	39	18 (46.20%)	0.651
Spironolactone (%)	78	33 (42.30%)	39	15 (38.50%)	39	18 (46.20%)	0.492
Hydrochlorothiazide (%)	78	0	39	0	39	0	\
Bumetanide (%)	78	0	39	0	39	0	\
Tolvaptan (%)	78	0	39	0	39	0	\
Hypertonic saline ^#^ (%)	78	0	39	0	39	0	\
K supplement (%)	78	62 (79.50%)	39	30 (76.90%)	39	32 (82.10%)	0.575
HA dosage (g)	39	30.00 (10.00–150.00) 42.56 ± 32.18	39	30.00 (10.00–150.00) 42.56 ± 32.18	NA	NA	\
Incidence of hyponatremia (%)	78	15 (19.20%)	39	3 (7.70%)	39	12 (30.80%)	0.010

Notes: * Data regarding use of paracentesis were extracted before the development of hyponatremia. ^#^ Data regarding use of hypertonic saline were extracted before the development of hyponatremia. Abbreviations: PSM: propensity score matching; Pts: patients; HA: human albumin; HBV: hepatitis B virus; HCV: hepatitis C virus; AUGIB: acute upper gastrointestinal bleeding, Hb: hemoglobin; WBC: white blood cell; PLT: platelet; TBIL: total bilirubin; ALB: albumin; ALT: alanine aminotransferase; AKP: alkaline phosphatase; Scr: serum creatinine; K: potassium; Na: sodium; PT: prothrombin time; INR: international normalized ratio; MELD: model for end-stage liver disease; NA: not applicable.

**Table 4 jcm-12-00107-t004:** *Long-term outcome* cohort—Characteristics of patients in the treatment study after PSM.

Variables	No. Pts	Overall	No. Pts	HA Group	No. Pts	Control Group	*p* Value
Age (years)	16	55.90 (32.78–80.79) 56.49 ± 13.32	8	56.69 (44.18–80.79) 58.80 ± 13.07	8	53.84 (32.78–70.79) 54.18 ± 14.04	0.529
Sex (male) (%)	16	10 (62.50%)	8	4 (50.00%)	8	6 (75.00%)	0.320
Etiology of liver cirrhosis							
HBV (%)	16	4 (25.00%)	8	3 (37.50%)	8	1 (12.50%)	0.248
HCV (%)	16	3 (18.80%)	8	1 (12.50%)	8	2 (25.00%)	0.522
Alcohol (%)	16	6 (37.50%)	8	3 (37.50%)	8	3 (37.50%)	1.000
Hypokalemia (%)	16	3 (18.80%)	8	3 (37.50%)	8	0	0.055
AUGIB (%)	16	10 (62.50%)	8	5 (62.50%)	8	5 (62.50%)	1.000
Infection (%)	16	3 (18.80%)	8	2 (25.00%)	8	1 (12.50%)	0.522
Ascites (%)	16	16 (100.00%)	8	8 (100.00%)	8	8 (100.00%)	\
Paracentesis * (%)	16	0	8	0	8	0	\
Laboratory tests							
Hb (g/L)	16	75.00 (59.00–135.00) 81.94 ± 22.29	8	74.5 (59.00–100.00) 77.13 ± 16.39	8	76.00 (60.00–135.00) 86.75 ± 27.26	0.563
WBC (10^9^/L)	16	6.45 (1.70–20.30) 7.16 ± 4.38	8	6.70 (2.30–20.30) 8.30 ± 5.70	8	6.25 (1.70–9.20) 6.03 ± 2.39	0.563
PLT (10^9^/L)	16	96.50 (22.00–215.00) 100.13 ± 59.72	8	82.50 (22.00–215.00) 83.50 ± 60.95	8	123.00 (30.00–203.00) 103.00 ± 67.30	0.128
TBIL (μmol/L)	16	34.50 (13.10–281.10) 59.68 ± 71.23	8	31.10 (13.10–177.90) 53.69 ± 54.44	8	36.45 (14.70–281.10) 65.68 ± 88.46	0.834
ALB (g/L)	16	28.05 (19.00–34.00) 27.04 ± 4.32	8	25.95 (19.00–28.50) 25.03 ± 3.37	8	30.00 (19.30–34.00) 29.05 ± 4.40	0.015
ALT (U/L)	16	24.04 (10.26–613.24) 68.07 ± 146.75	8	23.16 (10.26–68.00) 27.53 ± 19.09	8	40.66 (10.92–613.24) 108.61 ± 205.00	0.294
AKP (U/L)	16	85.12 (43.51–351.14) 114.80 ± 78.75	8	90.87 (43.51–187.00) 97.97 ± 47.90	8	85.12 (62.00–351.14) 131.64 ± 101.72	0.529
Scr (μmol/L)	16	66.99 (37.66–99.20) 67.65 ± 15.69	8	68.25 (37.66–90.10) 67.61 ± 17.79	8	66.99 (51.90–99.20) 67.69 ± 14.52	0.916
K (mmol/L)	16	3.94 (2.72–4.51) 3.77 ± 0.49	8	3.68 (2.72–4.12) 3.51 ± 0.53	8	4.01 (3.64–4.51) 4.04 ± 0.27	0.040
Na (mmol/L)	16	133.95 (127.00–134.90) 133.27 ± 2.14	8	133.20 (127.00–134.70) 132.31 ± 2.68	8	134.6 (132.80–134.90) 134.23 ± 0.73	0.073
PT (seconds)	16	16.75 (13.80–23.90) 17.78 ± 3.62	8	18.35 (13.80–23.90) 18.68 ± 4.27	8	16.05 (14.00–22.20) 16.89 ± 2.83	0.529
INR	16	1.38 (1.06–2.08) 1.49 ± 0.37	8	1.54 (1.09–2.08) 1.58 ± 0.42	8	1.33 (1.06–2.04) 1.40 ± 0.32	0.344
Child–Pugh score	16	9 (7–12) 9.44 ± 1.36	8	9 (8–12) 9.75 ± 1.49	8	9 (7–11) 9.13 ± 1.25	0.451
MELD score	16	9.65 (5.37–18.77) 10.65 ± 4.48	8	9.65 (5.37–18.77) 11.07 ± 4.95	8	9.75 (5.42–16.07) 10.23 ± 4.25	0.834
Treatments							
Desmopressin (%)	16	0	8	0	8	0	\
Terlipressin (%)	16	1 (6.30%)	8	0	8	1 (12.50%)	0.302
Furosemide (%)	16	9 (56.30%)	8	5 (62.50%)	8	4 (50.00%)	0.614
Torasemide (%)	16	11 (68.80%)	8	5 (62.50%)	8	6 (75.00%)	0.590
Spironolactone (%)	16	8 (50.00%)	8	3 (37.50%)	8	5 (62.50%)	0.317
Hydrochlorothiazide (%)	16	0	8	0	8	0	\
Bumetanide (%)	16	0	8	0	8	0	\
Tolvaptan (%)	16	0	8	0	8	0	\
Hypertonic saline ^#^ (%)	16	1 (12.50%)	8	1 (12.50%)	8	0	0.302
K supplement (%)	16	14 (87.50%)	8	7 (87.50%)	8	7 (87.50%)	1.000
HA dosage (g)	8	40.00 (20.00–180.00) 60.00 ± 53.72	8	40.00 (20.00–180.00) 60.00 ± 53.72	NA	NA	\
Improvement of hyponatremia (%)	16	10 (62.50%)	8	7 (87.50%)	8	3 (37.50%)	0.039

Notes: * Data regarding use of paracentesis were extracted after the development of hyponatremia. ^#^ Data regarding use of hypertonic saline were extracted after the development of hyponatremia. Abbreviations: PSM: propensity score matching; Pts: patients; HA: human albumin; HBV: hepatitis B virus; HCV: hepatitis C virus; AUGIB: acute upper gastrointestinal bleeding, Hb: hemoglobin; WBC: white blood cell; PLT: platelet; TBIL: total bilirubin; ALB: albumin; ALT: alanine aminotransferase; AKP: alkaline phosphatase; Scr: serum creatinine; K: potassium; Na: sodium; PT: prothrombin time; INR: international normalized ratio; MELD: model for end-stage liver disease; NA: not applicable.

## Data Availability

The datasets generated or analyzed during this study are available from the corresponding author on reasonable request.

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
