# Peer review of "Effects of Short-Term Human Albumin Infusion for the Prevention and Treatment of Hyponatremia in Patients with Liver Cirrhosis"

_jcm, 2022, doi:10.3390/jcm12010107_

Round 1

Reviewer 1 Report

This is an interesting work and the Authors are well prepared on the new finding of liver cirrhosis.

In this large retrospective study, the Authors compared development and improvement of hyponatremia between patients receiving Human albumin (HA) infusion during hospitalizations and did not, in order to evaluate the association of hyponatremia during hospitalization with the study outcomes in both Hospitalization Outcome cohort and Long-term Outcome cohort.

As resulted, in the two cohorts, HA infusion significantly decreased the incidence of hyponatremia and increased the rate of improvement of hyponatremia in cirrhotic patients during hospitalizations, which may influence the patients’ outcomes.

Methodologically the study is well structured, adequately appropriate and suitable to scientific rigour and ethical principles. The study design and material and methods are valid, both inclusion and exclusion criteria are compliant, as well as data collection; the statistical analysis seems valid; purposes and scientific contents are supported by valid References. The results are clear and the “Discussion” section is consistent with the results.

Minor problem:

1) Figure and tables are clear enough but may be improved for the editorial structure of the manuscript.

2) It would be useful to add a "Conclusion" section to reiterate the importance and usefulness of the study results: a "Take Home Message" for the Reader.

Reviewer question: Should patients continue HA infusion after discharge? In my opinion, this aspect could intrigue the Reader, therefore it should be developed and explained.

Apart from these, I believe that the manuscript is overall valid.

Author Response

Comment 1. Figure and tables are clear enough but may be improved for the editorial structure of the manuscript.

Reply 1. Thank you for your comments. We have improved the structure according to the requirement of Journal of Clinical Medicine. Please see the revised manuscript.

Comment 2. It would be useful to add a "Conclusion" section to reiterate the importance and usefulness of the study results: a "Take Home Message" for the Reader.

Reply 2. We have added the section of “Conclusion” according to your comments. Please see the words highlighted by yellow in Lines 410-414 in the revised manuscript.

Comment 3. Reviewer question: Should patients continue HA infusion after discharge? In my opinion, this aspect could intrigue the Reader, therefore it should be developed and explained.

Reply 3. Your comment is insightful. The HA infusion after discharge should be an important issue, which needs further investigation. In the In-hospital outcome cohort, we aimed to explore the effects of short-term HA infusion in the prevention and treatment of hyponatremia during hospitalizations and the role of the development and improvement of hyponatremia during hospitalizations for the in-hospital outcomes. In the Long-term outcome cohort, we aimed to explore the short-term HA infusion in the prevention and treatment of hyponatremia during hospitalizations and the role of the development and improvement of hyponatremia during hospitalizations for the long-term survival. Unfortunately, it should be acknowledged that the information regarding the use of HA infusion after discharge cannot be obtained in the Long-term outcome cohort, so this issue could not be clarified in our current study. Regardless, we have added this limitation in the section of limitations. Please see the words highlighted by yellow in Lines 405-407 in the revised manuscript.

Reviewer 2 Report

The manuscript presents a large scale of clinical investigation with clear objectives. The results and conclusions are convincing. The report will draw attention from clinicians and provide clinical advice regarding the use of albumin infusion for the prevention and treatment of hyponatremia in patients with liver cirrhosis.

Two suggestions are as the following.

     1.  In the discussion, provide more information regarding the 4 published studies using albumin infusion. For example, what are their conclusions?

          2.  feasible, in addition to the data analyses using combined male and female patient cohorts as presented by the manuscript, separate male and female patients and perform the same sets of data analyses. Sex is a primary variable in many diseases including liver disorders. It will be very interesting for clinicians and medical researches to see whether the prevention and treatment effects of albumin infusion on hyponatremia are sex-dependent. Although a lot of work needs to be done for data reorganization and analyses, it’s worth the effort.

Author Response

Comment 1. In the discussion, provide more information regarding the 4 published studies using albumin infusion. For example, what are their conclusions?

Reply 1. Thank you for your comment. We have added the descriptions of these 4 studies. Please see the words highlighted by yellow in Lines 343-353 in the revised manuscript.

Comment 2. feasible, in addition to the data analyses using combined male and female patient cohorts as presented by the manuscript, separate male and female patients and perform the same sets of data analyses. Sex is a primary variable in many diseases including liver disorders. It will be very interesting for clinicians and medical researches to see whether the prevention and treatment effects of albumin infusion on hyponatremia are sex-dependent. Although a lot of work needs to be done for data reorganization and analyses, it’s worth the effort.

Reply 2. Your comment is insightful. We have performed subgroup analyses according to the sex. Please see the results as follows. Additionally, the supplementary table regarding the subgroup analyses based on the sex were attached in the attachment..

Regardless of gender, the HA group had a significantly lower incidence of hyponatremia and higher rate of improvement of hyponatremia than the control group. Additionally, in both male and female groups, we further performed logistic regression analysis to explore the effect of HA infusion on the development and improvement of hyponatremia. The results still showed that HA infusion was significantly associated with decreased risk of developing hyponatremia during hospitalizations both in male (OR=0.306, 95%CI=0.187-0.502, P<0.001) and female (OR=0.288, 95%CI=0.150-0.553, P<0.001) patients, and increased the rate of improvement of hyponatremia during hospitalizations both in male (OR=4.193, 95%CI=2.484-7.076, P<0.001) and female (OR=6.979, 95%CI=2.378-20.480, P<0.001) patients.
